# Role of the YAP/TAZ-TEAD Transcriptional Complex in the Metabolic Control of TRAIL Sensitivity by the Mevalonate Pathway in Cancer Cells

**DOI:** 10.3390/cells12192370

**Published:** 2023-09-27

**Authors:** Younes El Yousfi, Rocío Mora-Molina, Abelardo López-Rivas, Rosario Yerbes

**Affiliations:** 1Centro Andaluz de Biología Molecular y Medicina Regenerativa-CABIMER, CSIC-Universidad de Sevilla-Universidad Pablo de Olavide, 41092 Seville, Spain; younes.elyousfi@cabimer.es (Y.E.Y.); rocio.mora@cabimer.es (R.M.-M.); abelardo.lopez@cabimer.es (A.L.-R.); 2Medical Physiology and Biophysics Department, Universidad de Sevilla and Instituto de Biomedicina de Sevilla (IBiS) (Hospital Universitario Virgen del Rocío/CSIC/Universidad de Sevilla), 41013 Seville, Spain

**Keywords:** statins, mevalonate pathway, TNF-related apoptosis-inducing ligand, YAP/TAZ-TEAD, cellular FLICE-like inhibitory protein

## Abstract

Different studies have reported that inhibiting the mevalonate pathway with statins may increase the sensitivity of cancer cells to tumor necrosis factor-related apoptosis-inducing ligand (TRAIL), although the signaling mechanism leading to this sensitization remains largely unknown. We investigated the role of the YAP (Yes-associated protein)/TAZ (transcriptional co-activator with PDZ-binding motif)-TEAD (TEA/ATTS domain) transcriptional complex in the metabolic control of TRAIL sensitivity by the mevalonate pathway. We show that depleting nuclear YAP/TAZ in tumor cells, either via treatment with statins or by silencing YAP/TAZ expression with siRNAs, facilitates the activation of apoptosis by TRAIL. Furthermore, the blockage of TEAD transcriptional activity either pharmacologically or through the ectopic expression of a disruptor of the YAP/TAZ interaction with TEAD transcription factors, overcomes the resistance of tumor cells to the induction of apoptosis by TRAIL. Our results show that the mevalonate pathway controls cellular the FLICE-inhibitory protein (cFLIP) expression in tumor cells. Importantly, inhibiting the YAP/TAZ-TEAD signaling pathway induces cFLIP down-regulation, leading to a marked sensitization of tumor cells to apoptosis induction by TRAIL. Our data suggest that a combined strategy of targeting TEAD activity and selectively activating apoptosis signaling by agonists of apoptotic TRAIL receptors could be explored as a potential therapeutic approach in cancer treatment.

## 1. Introduction

TRAIL is a member of the tumor necrosis factor (TNF) family that activates an apoptotic program in a broad variety of tumor cells upon binding to its pro-apoptotic receptors, sparing most normal cells [1,2]. The binding of TRAIL to its pro-apoptotic receptors induces the formation of the death-inducing signaling complex (DISC), recruiting the adapter protein Fas-associated death domain (FADD) and procaspase-8 [3,4]. Procaspase-8 is processed and activated to initiate an apoptotic cascade that results in cell death [3,4]. At the DISC, apoptosis signaling may be hindered by the caspase-8 inhibitor cFLIP [5]. The down-regulation of cFLIP levels is a frequent observation upon different treatments that sensitize tumor cells to TRAIL-induced apoptosis [6,7,8,9,10].

The mevalonate pathway is an essential metabolic pathway involved in the synthesis of different isoprenoids, including cholesterol, lipoproteins and vitamin D, among others [11]. Statins are inhibitors of the 3-OH-3-methyl-glutaryl CoA (HMG-CoA) reductase, a rate-limiting step in mevalonate synthesis [12,13]. Statins are the standard of care in the treatment of hypercholesterolemia and the prevention of cardiovascular pathologies [12,13]. In addition to reducing cholesterol levels, statins also affect the synthesis of other isoprenoids, essential constituents of signaling proteins implicated in cancer [14,15]. In this regard, different clinical studies have indicated the beneficial effect of statins in cancer patients both on their own and in combination with other anti-tumor treatments [16,17]. Interestingly, different works have shown that statins enhanced apoptosis activation by TRAIL, although the mechanism involved has not been revealed [18,19,20].

The Hippo pathway is an evolutionarily conserved protein kinase cascade that is involved in tumorigenesis and tumor progression [21,22]. YAP and TAZ transcriptional coactivators promote tissue proliferation and organ growth and are negatively regulated by the Hippo pathway [23]. Indeed, YAP has been shown to play a key role in inhibiting apoptosis in cancer cells [24,25,26,27], although a pro-apoptotic function of YAP has also been reported [28,29,30]. Importantly, several studies have demonstrated that YAP/TAZ nuclear localization and function can be controlled via the mevalonate pathway [31,32,33] through the geranylgeranylation and activation of RhoA GTPases [33].

We have examined the function of the YAP/TAZ-TEAD transcriptional system in the regulation of sensitivity to TRAIL via the mevalonate pathway in tumor cells. Herein, we report that inhibiting the mevalonate pathway using statin treatment markedly sensitized tumor cells to TRAIL. Sensitization to TRAIL through statins was abrogated in tumor cells overexpressing a constitutively active form of YAP. In this way, we show that both statins and YAP/TAZ knockdown reduced the cellular levels of the caspase-8 inhibitor cFLIP and sensitized cancer cells to a TRAIL-induced and mitochondria-regulated apoptotic process.

## 2. Materials and Methods

### 2.1. Cell Culture

HCT116 cells were kindly provided by Dr. J.A. Pintor-Toro (CABIMER, Seville, Spain). HCT116 Bax/Bak K.O. cells were a donation from Dr. C. Muñoz-Pinedo (Idibell, Barcelona, Spain). Both cell lines were maintained in McCoy’s medium containing 10% heat-inactivated fetal bovine serum, 2 mM L-glutamine, penicillin 50 U/mL and streptomycin 50 μg/mL. A549 cells were a gift from Dr. FJ. Oliver, Institute of Parasitology and Biomedicine CSIC, Granada, Spain. A549, MDA-MB468 (a gift by Dr. J. Arribas, VallD’Hebron Institute of Oncology, Barcelona, Spain) and HeLa cell lines were maintained in complete DMEM medium containing 10% fetal bovine serum, 2 mM L-glutamine, penicillin (50 U/mL) and streptomycin (50 μg/mL). Complete medium of A549 cells was also supplemented with glucose (4.5 g/L). All cell lines were regularly tested for mycoplasma contamination.

### 2.2. Reagents

Reagents for cell culture, buffer preparation and molecular biology were purchased from Merck/Sigma-Aldrich, (St. Louis, MO, USA). Simvastatin, cerivastatin, propidium iodide, puromycin, hygromycin, doxycycline and DAPI were purchased from Merck/Sigma-Aldrich. Geneticin (G-418) was from Gibco (Fisher Scientific, Loughborough, UK). XAV939 was obtained from Selleckchem (Houston, TX, USA). Human His-tagged recombinant TRAIL was produced as described [34].

### 2.3. Antibodies

Anti-Yap (SC-101199) and anti-GAPDH (SC-47724) antibodies were from Santa Cruz Biotechnology (Dallas, TX, USA). Anti-caspase 8 (1C12) (9746), was purchased from Cell Signaling Technology (Beverly, MA, USA). Anti-cFLIP monoclonal antibody 7F10 (ALX-804-961-0100) was from Enzo Life Sciences (Farmingdale, NY, USA). Horseradish peroxidase-conjugated secondary antibodies were purchased from DAKO (P0447, P0448, P0449) (Cambridge, UK). Alexa 488-conjugated secondary antibody was obtained from Jackson ImmunoResearch (Baltimore Pike, PA, USA).

### 2.4. Determination of Hypodiploid Apoptotic Cells

For the quantitative analysis of apoptosis, cells (3 × 10^5^/well) were cultured in 6-well plates and treated as shown in the figure legends. Cell cycle analysis (5 × 10^3^ cells/sample) was carried out in a FACSCalibur cytometer with Cell Quest software, Version 5.1 (Becton Dickinson, Mountain View, CA, USA), according to published procedures [35].

### 2.5. Analysis of Cell Viability

HeLa cells (3 × 10^5^/well) were cultured in 6-well plates and treated as shown in the figure legends. Cell viability was assessed after washing the cultures with phosphate-buffered saline (PBS)/0.1% bovine serum albumin and incubation for 15 min in the same buffer containing propidium iodide (2 μg/mL) in the dark. The uptake of propidium iodide was determined via flow cytometry with the Cell Quest Software.

### 2.6. Western Blotting of Proteins

After washing the cultures with PBS, cells (3 × 10^5^) were lysed in buffer containing 10 mM Na2HPO4, 3% SDS and 10% Glycerol. Once the protein concentration of lysates was measured with the Bradford reagent (Bio-Rad Laboratories, Hercules, CA, USA), Laemmli sample buffer was added and proteins were resolved through electrophoresis on SDS-polyacrylamide minigels. Proteins were identified via Western blotting as described [35], using GAPDH as a protein loading control.

### 2.7. Real-Time qPCR

Total RNA (1 μg) was isolated with PRImeZOL reagent (Canvax Biotech, Córdoba, Spain) and reverse-transcribed using an iScript cDNA synthesis kit (Bio-Rad, ref. 1708891). Real-time PCRs were carried out using SYBR green reagent (Bio-Rad, ref. 1725124). Alternatively, M-MLV reverse transcriptase (Invitrogen, Carlsbad, CA, USA) was used to reverse-transcribe total RNA, and mRNA expression was assessed via qPCR with primers and probes from Applied Biosystems (Waltham, MA, USA), normalized to hypoxanthine-guanine phosphoribosyltransferase levels. mRNA levels were assessed on the ABI Prism7500 sequence detection system.

Primers and probes for real-time qPCR:

Primers for SYBR green analysis:
GAPDHForward: 5′-ATGGGGAAGGTGAAGGTCG-3′ Reverse: 5′-GGGTCATTGATGGCAACAATATC-3′FLIP_L_Forward:5′-CCTAGGAATCTGCGTGATAATCGA-3′ Reverse: 5′-TGGGATATACCATGCATACTGAGATG-3′FLIP_S_Forward: 5′-GGGCCGAGGCAAGATAAGCAAGG-3′ Reverse: 5′-TCAGGACAATGGGCATAGGGTGT-3′CTGFForward: 5′-AGCTGACCTGGAAGAGAACA-3′ Reverse: 5′-CAGGCACAGGTCTTGATGAA-3′

Taqman primers and probes:
CYR61Hs00155479_m1HPRT1Hs01003267_m1

### 2.8. RNA Interference

siRNAs and non-targeting oligonucleotide were purchased from Sigma Aldrich (St. Louis, MO, USA). Dharmafect-1 (Dharmacon) was used for siRNA transfection, as described by the manufacturer. After 6 h, the transfection medium was replaced with regular medium and cells were further incubated for 48 h before analysis. The transfection of siRNAs in HCT116 cells was performed using jetPRIME (Polyplus transfection reagent), according to manufacturer guidelines. The transfection medium was removed, and cells were cultured in regular medium for 48 h before further analysis.

siRNAs sequences:
YAP#1:5′-GACAUCUUCUGGUCAGAGAdTdT-3′YAP#2:5′-CUGGUCAGAGAUACUUCUUdTdT-3′YAP#3:5′-GGUGAUACUAUCAACCAAAdTdT-3′TAZ#1:5′-ACGUUGACUUAGGAACUUUdTdT-3’TAZ#2:5´-AGGUACUUCCUCAAUCACAdTdT-3´cFLIP:5′-GGGACCUUCUGGAUAUUUUdTdT-3′Non-targeting control siRNA (Scr)5′-CUUUGGGUGAUCUACGUUAdTdT-3′

YAP/TAZ pairs of siRNAs:
siY/T#1: siYAP#1 + siTAZ#1 siY/T#2: siYAP#2 + siTAZ#2 siY/T#3: siYAP#3 + siTAZ#2

### 2.9. Retroviral and Lentiviral Vectors

pQCXIH-Myc-YAP-5SA was a gift from Kunliang Guan (Addgene plasmid # 33093; http://n2t.net/addgene:33093; accessed on 16 September 2019; RRID:Addgene_33093), FUW-tetO-wtYAP was a gift from Stefano Piccolo (Addgene plasmid # 84009; http://n2t.net/addgene:84009; accessed on 17 September 2019; RRID:Addgene_84009), FUW-tetO-EGFP was a gift from Stefano Piccolo (Addgene plasmid # 84041; http://n2t.net/addgene:84041; accessed on 17 September 2019; RRID:Addgene_84041), FUdeltaGW-rtTA was a gift from Konrad Hochedlinger (Addgene plasmid # 19780; http://n2t.net/addgene:19780; accessed on 17 September 2019; RRID:Addgene_19780) and pInducer20 EGFP-TEADi was a gift from Ramiro Iglesias-Bartolome (Addgene plasmid # 140145; http://n2t.net/addgene:140145; accessed on 4 September 2020; RRID:Addgene_140145). All these vectors for stable gene expression were obtained from the Addgene plasmid repository (Watertown, MA, USA). The pBabe puro vector has been described previously [35]. To perform silencing experiments, shRNAs against caspase-8 and TRAILR2, in a pSUPER vector (OligoEngine, Seattle, WA, USA), were digested and cloned between EcoR1 and Cla1 into a H1 promoter-driven GFP-encoding pLVTHM lentiviral vector [36].

shRNAs sequences:
TRAIL-R2:5′-GATCCCC**GACCCTTGTGCTCGTTGTC**TTCAAGAGA **GACAACGAGCACAAGGGTCT**TTTTTA-3′Caspase-8:5′-GATCCCC**GGAGCTGCTCTTCCGAATT**TTCAAGAG A**AATTCGGAAGAGCAGCTCC**TTTTTA-3′Scrambled (Scr):5′-GATCCCC**CTTTGGGTGATCTACGTTA**TTCAAGAGA **TAACGTAGATCACCCAAAG**TTTTTA-3′

To produce lentiviruses and retroviruses, HEK293-T cells were transfected with the corresponding vectors using the calcium phosphate method. After 48 h, lentivirus- or retrovirus-containing supernatants were collected, and viruses were concentrated via ultracentrifugation.

### 2.10. Generation of A549 Cell Lines

A549 cell lines were obtained after infection with lentiviruses or retroviruses, and selection was performed in medium with the corresponding antibiotic: puromycin (1.5 μg/mL), hygromycin (200 μg/mL) or G-418 (1 mg/mL). The infection of tumor cells with GFP-expressing lentiviruses was determined via flow cytometry.

### 2.11. Immunofluorescence

Cells were grown on coverslips and fixed for 10 min in 4% paraformaldehyde at room temperature. Following permeabilization with Triton X-100 (0.5%), cells on coverslips were incubated at room temperature for 1 h with primary antibodies diluted in blocking buffer. After washing with 0.1% PBS-Tween, cells were stained for 1 h with fluorescently labelled secondary antibody. Images were acquired with an Axio Imager 2 Zeiss microscope (Zeiss, Oberkochen, Germany). The nuclear localization of YAP was determined by calculating the ratio between nuclear YAP fluorescence and fluorescence in the cytoplasmic region immediately adjacent. Co-staining the nucleus with DAPI was used as a criteria to differentiate nuclear and cytoplasmic regions. Images analysis was performed using ImageJ software, Version 1.0.

### 2.12. Assessment of Luciferase Activity

YAP/TAZ-responsive reporter 8xGTII–luciferase plasmid (a gift from Stefano Piccolo, Addgene plasmid #34615; http://n2t.net/addgene:34615; accessed on 16 December 2020; RRID: Addgene_34615) was transfected in A549 cells together with a Renilla plasmid to normalize for transfection efficiency. Six hours after transfection, cells were treated with doxycycline (1 μg/mL, A549-iTEAD cells) or XAV939 (5 μM, A549 cells) during 24 h before collection. Luciferase activity in cell lysates was assessed using the Dual-Luciferase Reporter Assay System (Promega, Madison, WI, USA). Samples were analyzed in a Varioskan Flash microplate reader (Thermo Electron Corporation, Waltham, MA, USA). Every experimental condition was performed in duplicate.

### 2.13. Quantitative Analysis of Caspase-8 Activity

Cells growing in 6-well plates were treated as indicated in the figure legend. Following collection, cells were washed with PBS and resuspended in regular medium. After 90 min-incubation, caspase-8 activity was assessed using the Caspase-Glo^®®^8 Assay (Promega, Madison, WI, USA) by measuring the luminescence intensity in a Varioskan Flash microplate reader. Every experimental condition was performed in duplicate.

### 2.14. Global Protein Synthesis Determination by the SUnSET Method

General protein synthesis was assessed by the non-isotopic SUnSET assay as previously described [37]. Cells were pulse-labelled with puromycin and proteins in cell lysates were separated by denaturing electrophoresis. Rate of protein synthesis was determined via Western blot with an anti-puromycin antibody (clone 12d10, MABE343, Merck/Sigma Aldrich).

### 2.15. Statistics

Results are presented as the mean ± SD. Experiments were performed independently at least three times. GraphPAD Prism 8 software was used to carry out the statistical analysis (GraphPad Software, San Diego, CA, USA). The statistical tests used are indicated in the figure legends. *p* < 0.05 was considered significant. * *p* < 0.05; ** *p* < 0.01; *** *p* < 0.001; **** *p* < 0.0001.

## 3. Results

### 3.1. Control of TRAIL Sensitivity by the Mevalonate Pathway in Cancer Cells

Simvastatin, a mevalonate pathway inhibitor widely used in the treatment of cardiovascular disease, also shows inhibitory effects on various types of cancers [38]. Interestingly, different studies have described that inhibiting the mevalonate pathway with statins sensitizes tumor cells to apoptosis induction by death ligand TRAIL [18,19,20]. However, the signaling mechanism involved in this synergistic action remains poorly understood. To investigate the molecular mechanism responsible for the sensitization to TRAIL in tumor cells treated with statins, different tumor cell lines were incubated for 24 h with simvastatin followed by treatment with TRAIL for a further 24 h-period. In these cell lines, simvastatin significantly sensitized the cells to TRAIL-induced apoptosis (Figure 1A). Statins inhibit the enzymatic activity of HMG-CoA reductase, thus reducing the levels of the downstream metabolites of the mevalonate pathway [11]. To validate the on-target effects of simvastatin increasing the sensitivity of tumor cells to TRAIL, we carried out the rescue experiments via the supplementation of simvastatin-treated tumor cell cultures with mevalonate. Our results clearly indicated that simvastatin-mediated sensitization to TRAIL was abolished when cells were treated with the statin in the presence of mevalonate, prior to the addition of TRAIL (Figure 1A). These results were confirmed in experiments performed with cerivastatin, another inhibitor of HMG-CoA reductase (Figure 1B).

Collectively, our results demonstrate that the sensitivity of these tumor cells to the induction of the apoptotic pathway by TRAIL significantly depends on the levels of metabolites of the mevalonate pathway downstream of the activation of HMG-CoA reductase, the rate-limiting enzyme in the mevalonate pathway [11].

### 3.2. Metabolic Control of YAP/TAZ Subcellular Localization and Activity by the Mevalonate Pathway in Cancer Cells

Several studies have shown that the mevalonate pathway could be involved in the regulation of YAP/TAZ subcellular localization and activity in cancer cells [31,32,33]. Given the important role of YAP/TAZ in cancer cell survival [25,27], we examined whether YAP/TAZ may function as downstream effectors of the mevalonate pathway to promote resistance to TRAIL. First, we monitored by immunofluorescence the effect of simvastatin on the subcellular localization of YAP/TAZ in both the human lung adenocarcinoma cell line A549 and the triple-negative breast cancer cell line MDA-MB468 (Figure 2A). Interestingly, simvastatin treatment induced a significant cytoplasmic re-localization of YAP/TAZ in both cell lines (Figure 2A). Similar results were obtained by treating these cells with cerivastatin (Figure 2B). Importantly, the cytoplasmic re-localization of YAP/TAZ was not observed in cells treated with statins when mevalonate was added to the culture medium (Figure 2A,B). Moreover, we evaluated the effect of inhibiting the mevalonate pathway on CYR61 gene expression, a downstream target of the YAP/TAZ/TEAD transcriptional system [39]. Data shown in Figure 2C indicate that CYR61 gene expression was markedly impeded by simvastatin treatment and that mevalonate addition completely prevented the inhibition of YAP/TAZ/TEAD activity by the statin.

### 3.3. Inhibition of YAP/TAZ Signaling Mediates Statin-Induced Sensitization to TRAIL

The mevalonate pathway is known to be involved in the regulation of YAP/TAZ phosphorylation and activity independently of LATS1/2 kinases [33]. To determine whether the decreased YAP/TAZ signaling observed in cancer cells treated with statins plays a role in the sensitization to TRAIL-induced apoptosis, we first generated A549 cells expressing an activated form of YAP (YAP5SA), lacking inhibitory phosphorylation sites. As shown in Figure 3A, the expression of YAP5SA in A549 cells was sufficient to prevent simvastatin-induced sensitization to TRAIL.

To further demonstrate the role of YAP/TAZ proteins in the control of TRAIL sensitivity in cancer cells, we performed the simultaneous knockdown of YAP and TAZ proteins in the various cancer cell lines using two independent sets (siY/T#1 and siY/T#2) of previously validated YAP and TAZ siRNAs [40]. As shown in Figure 3B, YAP/TAZ knockdown significantly sensitized cancer cells to TRAIL-induced apoptosis. Importantly, sensitization to TRAIL by YAP/TAZ siRNAs was not observed in cells expressing a siRNA-insensitive form of wild-type YAP (Figure 3C).

### 3.4. Control of TRAIL Sensitivity by the YAP/TAZ-TEAD Signaling Module

Collectively, the above data suggest that transcriptional co-activators YAP/TAZ are involved in the regulation of TRAIL sensitivity by the mevalonate pathway in cancer cells. Nuclear YAP/TAZ mainly execute their transcriptional functions through their binding to members of the TEAD family of transcription factors [41,42]. To investigate the role of the YAP/TAZ-TEAD system in the control TRAIL sensitivity, we initially assessed the effect of the tankyrase inhibitor XAV939, a well-known inhibitor of TEAD activity [43], in the apoptotic response of cancer cells to TRAIL. As shown in Figure 4A, XAV939 treatment markedly decreased TEAD reporter activity in A549 cells. Interestingly, the incubation of A549 cells with XAV939 significantly sensitized these cells to TRAIL-induced apoptosis (Figure 4B).

To further demonstrate the role of the YAP/TAZ-TEAD signaling module in the regulation of TRAIL sensitivity in cancer cells, A549 cells expressing TEADi, an inhibitor of the interaction of YAP and TAZ with TEAD [44], were generated through infection with a tetracycline-inducible TEADi-GFP lentiviral vector (Figure 4C). The expression of TEADi in A549 cells markedly inhibited TEAD reporter activity (Figure 4D) and significantly sensitized these cells to TRAIL-induced apoptosis (Figure 4E).

### 3.5. YAP/TAZ-TEAD Signaling Confers Resistance to TRAIL-Induced Apoptosis by Controlling cFLIP Levels in Cancer Cells

Initiator caspase-8 is an essential component of the extrinsic pathway of apoptosis signaling induced upon death receptor activation by their cognate ligands [3,4,45] and cellular stress [46,47,48,49]. Furthermore, a growing body of evidences indicates that the activation of the mitochondria-operated pathway is a key event in the feedback amplification of apical apoptotic caspases by effector caspases [50]. To obtain further insight into the mechanism underlying the sensitization of cancer cells to TRAIL by statins, we first examined the role of the mitochondria in the induction of apoptosis by TRAIL upon YAP/TAZ knockdown. To this end, YAP/TAZ silencing was performed in HCT116 cells and their Bax/Bak deficient counterparts prior to treatment with TRAIL and apoptosis was then assessed. Results shown in Figure 5A illustrate that sensitization to TRAIL-induced apoptosis via YAP/TAZ knockdown was completely blocked in Bax/Bak KO cells. We next assessed the caspase-8 activation state in Bax/Bak KO cells, in which sensitization to TRAIL-induced apoptosis by YAP/TAZ silencing was completely inhibited. Interestingly, sensitization to TRAIL-induced caspase-8 processing (Figure 5B) and activation (Figure 5C) through YAP/TAZ knockdown were still observed in Bax/Bak KO cells, indicating that YAP/TAZ regulates early signaling in the extrinsic apoptotic pathway, although signal amplification through the mitochondrial pathway [50] is required to complete the apoptotic process.

The anti-apoptotic proteins cFLIPL and cFLIPS are key regulators of apical caspase-8 activation upon TRAIL receptor clustering by TRAIL [5,51,52] and cellular stress [53,54]. To further investigate the mechanism responsible for statin-mediated sensitization to TRAIL, we initially determined cFLIP levels in cells treated with simvastatin. As shown in Figure 6A, the treatment of A549 cells with simvastatin down-regulated the expression of both cFLIP_L_ and cFLIP_s_ proteins. Importantly, cFLIP loss in cells treated with simvastatin was prevented if mevalonate was added to the cell cultures (Figure 6A), further supporting the role of the mevalonate pathway in controlling the cellular levels of these anti-apoptotic proteins.

We also examined the role of the YAP/TAZ-TEAD module in the control of cFLIP expression in cancer cells by performing the genetic knockdown of YAP/TAZ using two different sets of siRNAs that efficiently silenced their expression in A549 and HCT116 cells (Figure 6B). Interestingly, YAP/TAZ knockdown resulted in a marked down-regulation of both cFLIP_L_ and cFLIPs expression levels in these cancer cells (Figure 6B). The role of the YAP/TAZ-TEAD signaling module in the control of cFLIP expression was also investigated with the tankyrase inhibitor XAV939 that efficiently reduced TEAD reporter activity in A549 cells (Figure 4A). Inhibiting TEAD activity with XAV939 resulted in a significant down-regulation of cFLIP proteins in these cancer cells (Figure 6C). The importance of YAP/TAZ-TEAD signaling in the control of cFLIP protein levels was further assessed in A549 cells expressing a tetracycline-inducible inhibitor of TEAD activity (TEADi-GFP). Results shown in Figure 6D demonstrate that inducing the expression of TEADi-GFP with doxycycline caused a marked loss of both cFLIPL and cFLIPs isoforms in A549 cells.

To investigate the mechanism underlying the observed down-regulation of cFLIP proteins levels upon the inhibition of the mevalonate pathway with simvastatin, we first assessed cFLIP mRNA levels in A549 tumor cells treated with the statin (Figure 6E) at a dose that induces maximal sensitization to TRAIL (Figure 1A). A significant decrease in the mRNA levels of both cFLIP isoforms was observed in tumor cells treated with simvastatin at a concentration that caused a marked inhibition of CTGF gene expression (Figure 6E), a downstream target of the YAP/TAZ/TEAD transcriptional module. Additionally, simvastatin has been reported to repress protein synthesis by reducing the cellular levels of eukaryotic initiation factor 2B [55]. Cellular FLIP isoforms are short-lived proteins that can be degraded by the proteasome after ubiquitination [56]. Consequently, a reduction in the global rate of protein synthesis following statins treatment could result in cFLIP loss through proteasomal degradation, and this modulation may lead to TRAIL sensitization. Therefore, we first assessed the rate of global protein synthesis inhibition upon statin treatment. Data shown in Figure 6F demonstrate that protein synthesis was reduced in tumor cells treated with simvastatin and restored in the presence of mevalonate (Figure 6F), which closely correlated with the regulation of FLIP levels in these cells by the mevalonate pathway. Global protein synthesis was also attenuated following YAP/TAZ knockdown in A549 cells (Figure 6F), which may also explain the down-regulation of cFLIP expression observed under these conditions (Figure 6B). To demonstrate the importance of the loss of cFLIP proteins observed in cells treated with simvastatin or by targeting YAP/TAZ-TEAD signaling in the sensitization to TRAIL-induced apoptosis, we depleted both cFLIP isoforms in A549 cancer cells with a siRNA oligonucleotide set prior to treatment with TRAIL. As shown in Figure 6G, the simultaneous silencing of the two cFLIP isoforms markedly sensitized these cancer cells to TRAIL-induced apoptosis.

Taken together, the above results suggest that the metabolic control of YAP/TAZ-TEAD transcriptional activity by the mevalonate pathway may represent an important mechanism controlling the expression of cFLIP (Figure 7), a key modulator of caspase-8 activity, and thus regulates the response of cancer cells to the clustering of proapoptotic receptors by death ligand TRAIL. We cannot rule out that changes in the expression levels of regulators of the bcl-2 family by statins may also be involved in facilitating the activation of the mitochondrial pathway of apoptosis upon TRAIL receptor activation by its ligand.

## 4. Discussion

TRAIL has long been reported to be a potent inducer of apoptosis in tumor cells, with non-transformed cells being more resistant to the pro-apoptotic action of this cell death ligand [1,2]. Based on these preclinical observations, different clinical trials on the use of TRAIL and agonistic TRAIL receptor antibodies for human cancer therapy have been undertaken [57]. However, primary tumors are normally resistant to TRAIL [58], and these findings may explain why clinical trials conducted so far have not achieved the expected results [57,59]. Therefore, deciphering the molecular determinants and signaling pathways responsible for TRAIL resistance should provide new targets for therapeutic intervention. In this way, sensitizing resistant tumor cells to TRAIL-induced apoptosis through different strategies would increase the therapeutic potential of TRAIL.

The metabolic control of the sensitivity of tumor cells to TRAIL-induced apoptosis has been the subject of different studies [8,60,61]. In this regard, the role of the mevalonate pathway in the control of the apoptotic response of tumor cells to TRAIL has been previously reported [18,19,20], although the mechanism involved has not yet been revealed. In our study, we demonstrate that inhibiting with statins the activity of HMG-CoA reductase, the rate-limiting enzyme in the mevalonate pathway [12,13], down-regulates cFLIP levels and facilitates the activation of an apoptotic process by TRAIL in different human cancer cell lines, by hindering the nuclear localization of the transcriptional coactivators YAP and TAZ.

The decrease in cFLIP_L_ levels upon statin treatment observed in our work could result in an elevation of the caspase-8/cFLIP_L_ ratio in the DISC of statin-treated cells, which should facilitate caspase-8 activation and promote apoptosis [52,62]. In this regard, it has been shown that different treatments inducing the down-regulation of cFLIP frequently lead to subsequent sensitization to TRAIL-induced apoptosis [10,63,64,65]. Moreover, the pivotal role of cFLIP down-regulation in TRAILR2-mediated apoptosis has been also demonstrated in tumor cells undergoing endoplasmic reticulum [53] or metabolic stress [54]. Importantly, elevated levels of cFLIP isoforms have been reported in human tumor samples from different cancers [66], suggesting a pro-tumoral role of this inhibitor of the extrinsic apoptotic pathway. Therefore, inhibiting cFLIP function by targeting the Hippo pathway downstream effectors YAP/TAZ may represent a potential antitumor strategy [67].

In addition to reducing the expression of a key regulator of the extrinsic apoptotic pathway like cFLIP, the inhibition of the mevalonate pathway via the treatment of cancer cells with statins may also promote the TRAIL-induced activation of apoptosis by enhancing the expression of apoptotic TRAIL receptors in glioblastoma and prostate cancer cells [20,68]. Furthermore, by lowering the levels of Bcl-2 and up-regulating pro-apoptotic Bax expression in cancer cells, statins may enable the activation of the intrinsic pathway of apoptosis [69], thus facilitating the amplification of apoptosis signaling through the mitochondria [50].

Agonistic antibodies against proapoptotic TRAIL receptors and non-tagged versions of recombinant TRAIL have been tested in clinical trials of various malignancies [57]. Unfortunately, despite broad tolerability, the clinical benefit of these TRAIL-based monotherapies has been rather limited, likely due, among other reasons, to the intrinsic resistance of most primary human tumors and the low agonistic activity of these agents [59]. Therefore, second-generation formulations of TRAIL and agonistic antibodies have been developed to increase the agonistic bioactivity against tumor cells [59,70]. In this context, our data suggest that the pharmacologic targeting of YAP/TAZ-TEAD transcriptional signaling with benzoporphyrin derivatives or allosteric inhibitors [71,72] may synergize with agonists of apoptotic TRAIL receptors to overcome the TRAIL resistance of tumor cells. Collectively, our results on the regulation of the YAP/TAZ-TEAD system by the mevalonate pathway in tumor cells warrant further studies to ascertain the feasibility of inhibiting this pathway with statins for therapeutic intervention in cancer, in combination with novel agonists of pro-apoptotic TRAIL receptors.

## 5. Conclusions

Since the initial reports of the pro-apoptotic effect of TRAIL on tumor cells and its lack of toxicity in clinical trials, TRAIL has been considered a potential antitumor therapy. However, the various clinical trials conducted using TRAIL as a monotherapy have not confirmed these encouraging data. Therefore, it has become essential to find treatments that can sensitize tumor cells to the pro-apoptotic action of TRAIL without increasing the toxicity of these combined therapies. Our results have shown that the inhibition of the metabolic pathway of mevalonate by statins in various tumor cell lines facilitates the activation of the extrinsic pathway of apoptosis by TRAIL and the death of these tumor cells. Our studies also indicate that this sensitizing action of statins is due to their inhibitory effect on the YAP/TAZ-TEAD transcriptional module. In conclusion, our results suggest that modulating the activity of the Hippo pathway may represent a therapeutic strategy that overcomes the resistance of tumor cells to TRAIL. However, preclinical work should be performed to confirm this hypothesis before its application as an antitumor therapy in humans could be considered.

## Figures and Tables

**Figure 1 cells-12-02370-f001:**
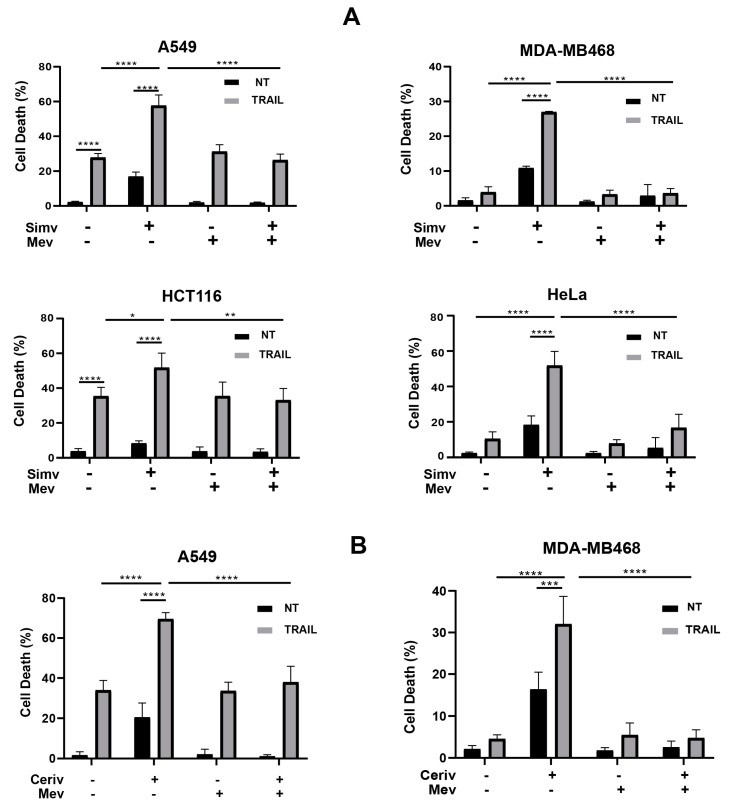
Inhibition of the mevalonate pathway by statins sensitizes cancer cells to TRAIL. Apoptosis in cancer cells treated with (**A**) simvastatin (Simv, 5 μM) or (**B**) cerivastatin (Ceriv, 1 μM), in the presence or absence of mevalonate (Mev, 0.5 mM) during 24 h prior to incubation with or without TRAIL. Cell death was determined via the quantification of hypodiploid cells through flow cytometry (5 × 10^3^ cells/sample) (A549, MDA-MB468, HCT116). In HeLa cells, cell viability was assessed via propidium iodide uptake. Data show the mean ± SD of at least three independent experiments. * *p* < 0.05; ** *p* < 0.01; *** *p* < 0.001; **** *p* < 0.0001. Two-way ANOVA with Tukey’s multi-comparisons test.

**Figure 2 cells-12-02370-f002:**
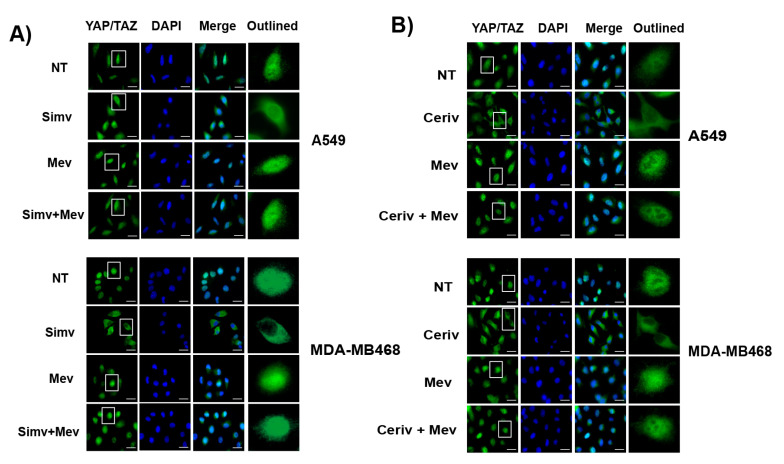
Nuclear localization and activity of YAP/TAZ are regulated by the mevalonate pathway in cancer cells. (**A**) Representative images of YAP/TAZ immunofluorescence (left panel) in cells treated with dimethylsulphoxide (NT) or simvastatin (Simv) alone or with mevalonate (Mev) for 24 h. Quantification of nuclear/cytosolic YAP/TAZ (right panel) from three independent experiments. (**B**) Representative images of YAP/TAZ immunofluorescence (left panel) in cells treated with dimethylsulphoxide (NT) or cerivastatin (Ceriv) alone or with mevalonate (Mev) for 24 h. Quantification of nuclear/cytosolic YAP/TAZ (right panel) from three independent experiments. At least 100 cells were analyzed in every experiment. Scale bars, 25 μm. (**C**) A549 cells were cultured with dimethylsulphoxide (NT) or simvastatin (Simv) alone or with mevalonate (Mev) for 24 h, and CYR61 mRNA levels were assessed using RT-qPCR. Data show the mean ± SD of at least three independent experiments. * *p* < 0.05; ** *p* < 0.01; *** *p* < 0.001; **** *p* < 0.0001. Two-way ANOVA with Tukey’s multi-comparisons test.

**Figure 3 cells-12-02370-f003:**
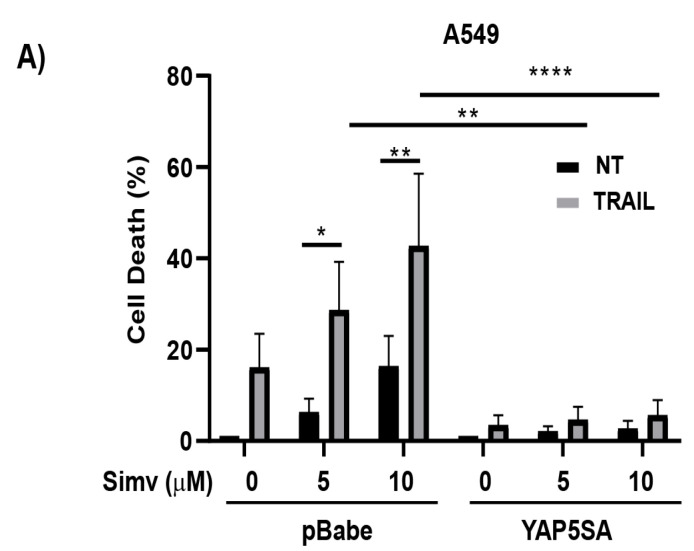
Role of YAP/TAZ in the control of TRAIL sensitivity via statins. (**A**) A549-pBabe or A549-YAP5SA cells were treated for 24 h with the indicated doses of simvastatin prior to incubation in the presence or absence of TRAIL for 24h. Cell death was analyzed by subG1 quantification (5 × 10^3^ cells/sample). Data are normalized to NT. (**B**) Cells were transfected with two different siRNA oligonucleotides pairs targeting both YAP and TAZ, as described in the Materials and Methods section. A scrambled RNA was also used as non-targeting control oligonucleotide. After 30 h, cells were treated with TRAIL for 24 h. Cell death was analyzed via subG1 quantification. (**C**) A549-EGFP or A549-wtYAP cells were transfected with siRNAs against both YAP and TAZ (YAP/TAZ#1). After 6 h, cells were treated with doxycycline (1 μg/mL). At 30 h post-transfection, cells were treated with TRAIL for 24 h and apoptotic cell death was then determined. Protein knockdown was confirmed via immunoblot analysis. Data show the mean ± SD of at least three independent experiments. * *p* < 0.05; ** *p* < 0.01; *** *p* < 0.001; **** *p* < 0.0001. Two-way ANOVA with Tukey’s multi-comparisons test.

**Figure 4 cells-12-02370-f004:**
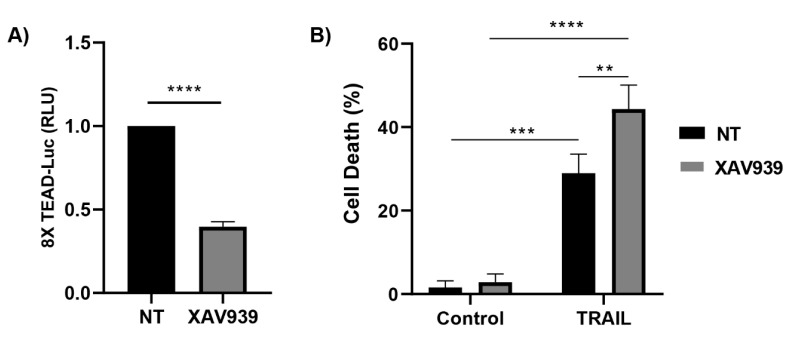
Regulation of sensitivity of cancer cells to TRAIL by the YAP/TAZ-TEAD signaling module. (**A**) A549 cells were cultured in the presence or absence of XAV939 5 μM for 24 h and transcriptional activity of TEAD was determined with a luciferase reporter that contains tandem TEAD-binding sites. (**B**) A549 cells were treated as in (**A**) before adding TRAIL for 24 h. Cell death was assessed via subG1 quantification (5 × 10^3^ cells/sample). (**C**) Microscope images showing GFP expression in iTEAD cells after doxycycline treatment (1 μg/mL, 24 h). (**D**) TEAD transcriptional activity measured via luciferase assay in A549 iTEAD cells cultured for 24 h in the presence or absence of doxycycline (1 μg/mL). (**E**) A549 iTEAD cells were treated with Doxycycline for 24 h before TRAIL treatment for 24 h. Cell death was measured as a quantification of subG1 population. Data show the mean ± SD of at least three independent experiments. ** *p* < 0.01; *** *p* < 0.001; **** *p* < 0.0001, unpaired *t*-test (**A**,**D**). Two-way ANOVA with Tukey’s multi-comparisons test (**B**,**E**).

**Figure 5 cells-12-02370-f005:**
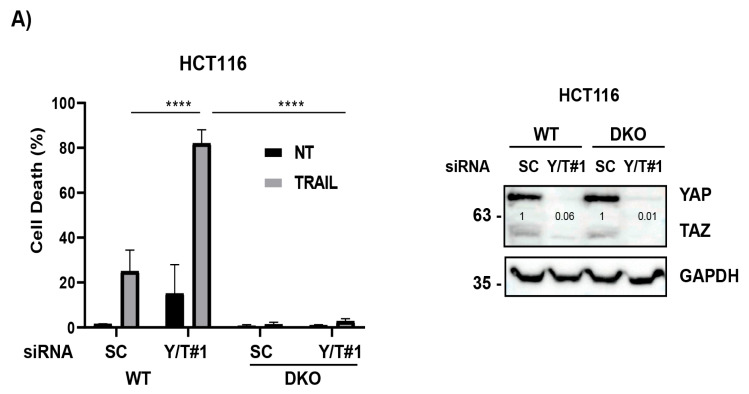
YAP/TAZ knockdown facilitates the TRAIL-induced activation of apical caspase-8 in cancer cells. (**A**) Wild type (WT) or Bax/Bak DKO HCT116 cells were transfected with siRNA oligonucleotides targeting both YAP and TAZ (YAP/TAZ#1). A scrambled RNA oligonucleotide was also used as non-targeting control oligonucleotide. After 30 h, cells were treated with TRAIL for 24 h and apoptosis determined via subG1 analysis (5 × 10^3^ cells/sample). Protein knockdown was confirmed via immunoblot analysis. (**B**) HCT116 Bax/Bak DKO cells were transfected as in (**A**) and then treated with TRAIL (50 ng/mL) for the indicated times. Caspase-8 processing was analyzed via Western blotting. (**C**) HCT116 Bax/Bak DKO cells were transfected as in (**A**) and then treated for 2 h with TRAIL. Caspase-8 activity was determined as described in the Materials and Methods section. Data show the mean ± SD from three independent experiments. **** *p* < 0.0001. Two-way ANOVA with Tukey’s multi-comparisons test.

**Figure 6 cells-12-02370-f006:**
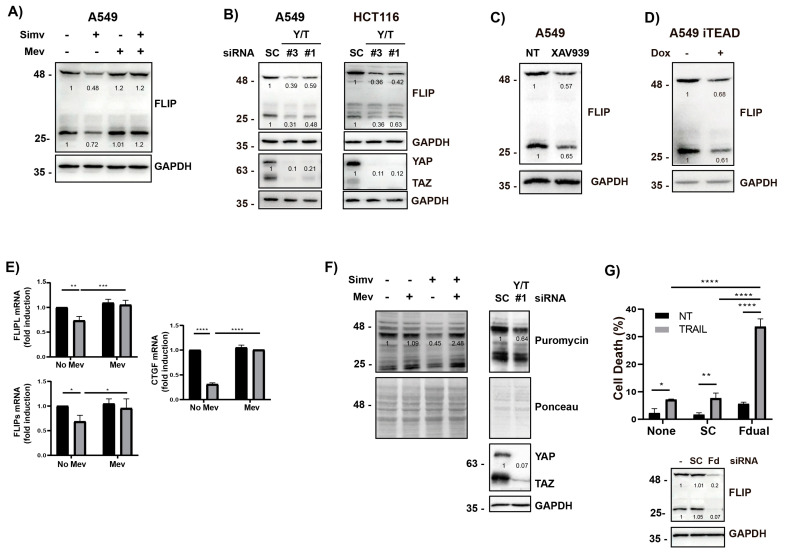
Control of cFLIP levels in cancer cells by the YAP/TAZ-TEAD signaling module. (**A**) A549 cells were treated with simvastatin (Simv, 5 μM) in the presence or absence of mevalonate (Mev, 0.5 mM) for 30 h, and cFLIP levels were analyzed via Western blotting. (**B**) Cells were transfected with siRNA oligonucleotides targeting both YAP and TAZ (#1 and #3). A scrambled RNA was also used as non-targeting control oligonucleotide (SC). After 30 h, cFLIP levels were analyzed via Western blotting. (**C**) A549 cells were treated with XAV939 5 μM for 24 h, and cFLIP levels cells were analyzed via Western blotting. (**D**) A549 iTEAD cells were treated with doxycycline for 24 h and cFLIP levels cells were analyzed via Western blotting. (**E**) A549 cells were treated as in (**A**) and mRNA levels of both cFLIP isoforms, and CTGF were determined via real time-qPCR. (**F**) A549 cells were either treated as in (**A**) or transfected with siRNA oligonucleotides targeting YAP/TAZ (YT#1), as indicated in (**B**). Puromycin (1 μg/mL) was added for the last 10 min of incubation. The incorporation of puromycin into cellular proteins during the 10 min pulse was determined via Western blotting with an anti-puromycin antibody, as described in the Materials and Methods section. (**G**) A549 cells were transfected with siRNA oligonucleotides targeting both cFLIP isoforms (Fd). After 6 h, cells were treated with TRAIL for 24 h. Cell death was analyzed via subG1 quantification (5 × 10^3^ cells/sample). Protein knockdown was assessed via immunoblot analysis. Data show the mean ± SD of at least three independent experiments. * *p* < 0.05; ** *p* < 0.01; *** *p* < 0.001; **** *p* < 0.0001. Two-way ANOVA with Tukey’s multi-comparisons test.

**Figure 7 cells-12-02370-f007:**
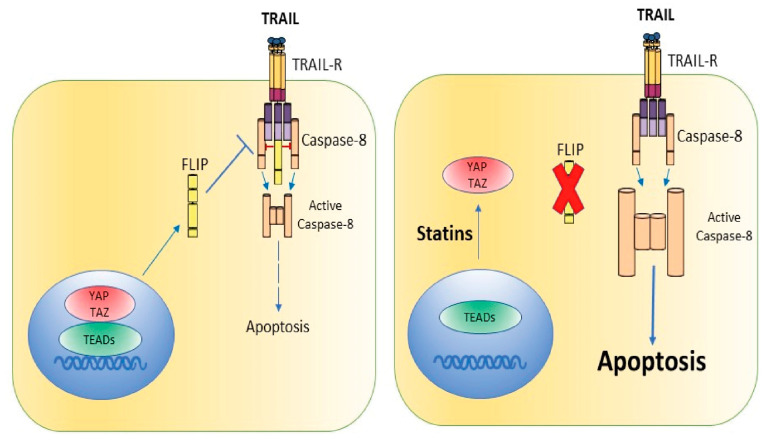
Schematic overview of our current data suggesting the mechanism of statin-induced sensitization to TRAIL-induced apoptosis in cancer cells.

## Data Availability

All data generated or analyzed during this study are included in the main text.

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
