# Peer review of "Role of the YAP/TAZ-TEAD Transcriptional Complex in the Metabolic Control of TRAIL Sensitivity by the Mevalonate Pathway in Cancer Cells"

_cells, 2023, doi:10.3390/cells12192370_

Round 1

Reviewer 1 Report

This study reports that HMG-CoA reductase inhibitors statins sensitize tumor cells to TRAIL-induced apoptosis via suppressing YAP/TAZ-TEAD transcription activity. Authors employed various genetic and pharmacological perturbations to evaluate the role of HMG-CoA reductase and mevalonate pathway in TRAIL-sensitivity, the role of YAP/TAZ-TEAD in mediating the tumor resistance to TRAIL-induced apoptosis, and the effect of statins on YAP/TAZ-TEAD activity and tumor cell sensitivity to TRAIL-induced apoptosis. Overall the study was well designed and the result is interesting. Listed below are minor comments to improve the quality of the paper.

1. In line 355-358, it is described that Simvastatin down-regulated cFLIP and this effect was prevented by mevalonate. But Fig. 6A shows an opposite result. It looks like the Simvastatin and Mevalonate are mislabeled.

2. Authors need to interpret why CTGF mRNA was tested in Fig. 6E and what the result tells.

3. Line 353, the sentence is not complete. Need to revise.

Reviewer 2 Report

In this work, El Yousfi and coworkers aim to elucidate the mechanistic insights by which the mevalonate pathway regulates the control of apoptotic response of tumor cells to TRAIL. In particular, the authors demonstrate the key role of YAP/TAZ in the regulation of this system, then adding important information that may help to ascertain the molecular mechanisms responsible for the sensitization to TRAIL in tumor cells.

The article is well-written, the figures are nicely presented and the data using a large battery of cell lines, different methodological approaches (genetic or pharmacologic depletion) and performing the right experimental controls clearly support the conclusions. The statistical analysis is properly applied.

Major revision:

1) Could the authors show in Figure 2 the images single treated with Mev?

Some minor comments are suggested to improve the edition of the manuscript:

1) Why YAP is sometimes named as Yap1 or Yap and other times as YAP.

2) Sometimes the size of the letters throughout the text is different. Please check.

3) Sometimes micrograms appears as ug or mg. Please use Greek symbol.

4) Sometimes minutes are defined as min or minutes. In this regard, the description of panels in the figure legends also differs among them (i.e Fig. 6). Please homogenize.

5) Please indicate the number of cells quantified by flow cytometry in both the M&M section and the corresponding Figure legends.

6) Please indicate the molecular weight of proteins detected by WB in Fig  3C.

7) Please homogenize the representation of bar charts (i.e Fig 6E and 6G).

8) Since the authors quantify protein expression levels in most of WB, I would suggest to show this quantification in all the blots.

9) What is the meaning of “Table 8” in line 353?).

10) Scale bars are not clearly visualized in the panels.
